# The Israeli Phage Bank (IPB)

**DOI:** 10.3390/antibiotics9050269

**Published:** 2020-05-21

**Authors:** Ortal Yerushalmy, Leron Khalifa, Naama Gold, Chani Rakov, Sivan Alkalay-Oren, Karen Adler, Shira Ben-Porat, Reut Kraitman, Niv Gronovich, Kerem Shulamit Ginat, Mohanad Abdalrhman, Shunit Coppenhagen-Glazer, Ran Nir-Paz, Ronen Hazan

**Affiliations:** 1Institute of Dental Sciences, School of Dentistry, Hebrew University of Jerusalem, Jerusalem 91120, Israel; ortal.yerushalmy@mail.huji.ac.il (O.Y.); leronK@ekmd.huji.ac.il (L.K.); naamaemmagold@gmail.com (N.G.); chani2061@gmail.com (C.R.); siavan.alkalay@mail.huji.ac.il (S.A.-O.); karen.adler@mail.huji.ac.il (K.A.); shira.benporat@mail.huji.ac.il (S.B.-P.); reutkraitman@gmail.com (R.K.); shunitc@ekmd.huji.ac.il (S.C.-G.); 2Alpha Program, Future Scientists Center for the Advancement of the Gifted and Talented, Jerusalem 9190401, Israel; niv7804@gmail.com (N.G.); kerem1ginat@gmail.com (K.S.G.); 3Hadassah Medical Center, Department of Infectious Diseases, Jerusalem 91120, Israel; muhaned@hadassah.org.il (M.A.); nirpaz@hadassah.org.il (R.N.-P.)

**Keywords:** phage therapy, phage bank, phage database, bacteriophage, IBP

## Abstract

A key element in phage therapy is the establishment of large phage collections, termed herein “banks”, where many well-characterized phages, ready to be used in the clinic, are stored. These phage banks serve for both research and clinical purposes. Phage banks are also a key element in clinical phage microbiology, the prior treatment matching of phages and antibiotics to specific bacterial targets. A worldwide network of phage banks can promote a phage-based solution for any isolated bacteria. Herein, we describe the Israeli Phage Bank (IPB) established in the Hebrew University, Jerusalem, which currently has over 300 phages matching 16 bacteria, mainly pathogens. The phage bank is constantly isolating new phages and developing methods for phage isolation and characterization. The information on the phages and bacteria stored in the bank is available online.

## 1. Introduction

The high specificity of bacteriophages (phages) to their target is a double-edged sword. On the one hand, in contrast to most antibiotics, it allows targeted treatment without causing dysbiosis to the microbiome [1,2,3]. On the other hand, it limits the flexibility of using phages clinically because an efficient treatment requires an almost perfect match of phages to their bacterial target. Subsequently, several attempts to use generic phage cocktails have failed [4,5]. Thus, prior to treatment, a quick and efficient series of tests, termed “clinical phage microbiology”, need to be taken to match the best combination of phages and antibiotics to the target bacteria, a concept that is reviewed elsewhere (Gelman et al., submitted). One of the most important key elements in matching a phage to its target bacteria is a large collection of phages, termed herein “phage bank”. This bank should contain phages that are characterized and ready to be used in a clinical setting against as many strains of bacteria as possible [6]. The phages in the bank can be used for research, diagnostics and treatments, as well as for many other phage-based applications. 

In comparison to other biological collections {Merck, ATCC, ECACC and others}, phage banks are currently less common and as a result, the process of finding and obtaining phages for research and, mainly, treatment is far from reaching its high potential. In addition, often the information in the existing banks and collections is insufficient. Among the current collections, perhaps the largest and most significant resources of phages for researchers are the well-known ATCC (https://www.atcc.org) and the Public Health England (PHE) collections, which include the NCTC (https://www.phe-culturecollections.org.uk). Both have a variety of phages for many bacterial hosts. One noteworthy initiative is the Phage Directory (https://phage.directory) which aims to establish a phage-related network by publishing phage-related issues and connecting demands for phages, mainly for treatment, with phage-related labs and collections. Recently, Phage Directory has reviewed several phage collections (https://phage.directory/capsid/phage-banks) in its blog “Capsid and Tail” (https://phage.directory/capsid). Table 1 summarizes an updated list of the main phage collections and banks worldwide. Nevertheless, most collections publish only limited information about their phages, mainly the target bacterium and more rarely the genome sequence of the phage.

Over the past two years, our team from the Hebrew University and Hadassah Medical Center in Jerusalem has been working on advancing phage therapy in Israel [7]. So far, we have treated four patients locally, and sent several phages to other centers for treatments. 

As part of this initiative, we have set up the Israeli Phage Bank (IPB). This bank’s goal is to collect phages against a wide range of bacteria, pathogens, and nonpathogens, and to play a major role in the establishment of a phage therapy center that will provide phages, treatment, and other phage-related services (Figure 1).

## 2. Methods and Results

### 2.1. Bacteria Collection

As a first step towards establishing the bank, we have prepared a collection of bacteria that will serve as the target hosts. This collection is constantly being expanded with new bacteria (Table 2). Most of the bacterial strains are clinical isolates collected mainly from the clinical laboratory at Hadassah Medical Center in Jerusalem. In addition, our collection contains a reference to bacterial strains from the ATCC and other sources. To date (April 2020), our bacterial collection harbors 382 strains of 21 species (Table 2), including opportunistic pathogens, environmental germs and plant pathogens. Each bacterium in the collection is kept frozen at −80 °C and its growing conditions’ protocol is recorded. Basic characterization was done for all bacteria including determination of sensitivity to antibiotics and known phages. 

### 2.2. Phage Isolation

For phage isolation, we employ both well-known protocols [8,9] and unique protocols that we developed for high-throughput screening of large samples, such as large water bodies and sewage. These protocols will be detailed elsewhere. We are constantly optimizing our isolation protocols according to the type of the target bacteria and the type and size of the environmental samples. The sources of the samples include urban sewage, dental sewage, feces and water from a zoo, soil and water from nature reserves, hospitals, public areas, water bodies, and more. Unused portions of the samples are stored at 4 °C, fecal samples for up to two months and environmental samples for longer, to be used in future screening. Routinely, prior to the test on bacteria, the samples are enriched by the addition of the target bacteria, incubated and if needed, concentrated by ultra-centrifugation. Potential phages are identified by plaque assay or by lysis observed using a 96-well plate reader [10]. Potential phages are validated by several transfers of plaques to new liquid or solid cultures of their bacterial target. Propagation of the number of plaques or an increase in lysis parameters in liquid indicates the existence of phages, while dissolving of the plaques or a decrease in lysis suggests the presence of a toxic material that harms the bacteria rather than a multiplying phage. For storage, phages are grown until maximal lysis is achieved and frozen in 25% glycerol at −80 °C. Alternatively, phages are incubated with their target for 20 min to achieve absorbance and then the coculture is frozen.

### 2.3. Phage Characterization

Each isolated phage is characterized for its efficacy against the target bacteria using several growth methods in liquid and on agar plates [10]. In addition, we aimed to fully sequence and analyze the genome of each phage. The sequence analysis includes annotations, search for virulence factors, and determination of phylogeny and similarity to close phages. An example would be the characterization we performed on phages Granit and CHEMY of *Klebsiella pneumoniae* [11]. Lysogeny is suggested by the BLAST search of the phage genome in bacterial genomes. Phages that do not show homology to prophages in the bacterial genome are considered lytic. Transmission electron microscopy (TEM) determines phage morphology and supports its phylogeny identification [10]. Characterization also includes determination of growth curve patterns using scoring methods based on logistic equations and different plaque-dependent morphologies, efficiency against biofilm (when relevant), using ex vivo models and more [10,12]. Last, the phage range of infectivity is tested against various isolates from the collection [10].

### 2.4. IBP Phage Collection

As of April 2020, the IPB contains 289 phages against 21 different bacteria (Table 3), including the major pathogens *Staphylococcus aureus*, *Klebsiella pneumonia*, *Pseudomonas aeruginosa*, *Enterococcus faecium*, *Enterococcus faecalis*, *Acinetobacter baumannii*, and *Escherichia coli*. Most of them were isolated by us; several, such as *E. coli* T4, T7 and lambda, were purchased from the ATCC and a few were obtained from other labs and companies. Noteworthy are KpKT21-φ-I and AbKT21-φ-III, which were kindly obtained from Adaptive Phage Therapy (APT) and were used for the first in-human treatment in Israel [7]. The phages are in various steps of characterization and so far, the genomes of 58 were sequenced. 

### 2.5. The Phage Database

More information about the phage collection is currently available on our web page (https://ronenhazanlab.wixsite.com/hazanlab/the-404-israeli-phage-bank), which is constantly being updated. For each phage, we intend to have links to its genome sequence and papers, if mentioned. The database will include more information such as plaque morphology, rate of growth on the different strains, interactions with antibiotics, TEM pictures, and the phylogenetic data.

### 2.6. Bank Activities

So far, seven of our phages have been sent to several laboratories around the world as part of collaborations for research and treatment purposes, and three of them already have been used to treat humans in Israel, the USA, Germany, and Australia (to be published elsewhere). Additional phages were tested or are expected to be tested in animal trials. 

One important goal of the bank is to achieve full coverage of pathogen collections with the idea that if we have full phage coverage, for instance, as in one of our significant projects where we screen phages against ~500 XDR P. aeruginosa strains collected in Hadassah Medical center over more than 30 years, the chances of a new strain to emerge and be resistant to all phages is quite low. Furthermore, succeeding to provide such coverage gives us the confidence that we will also succeed in finding phages against new resistant isolates, should they emerge. 

Yet another project in the bank is developing “training” and improving methods to quickly adapt existing phages to a new bacterial strain, when their efficiency is low [13,14]. To this end, we are “training” the phages using Darwinian mutation-selection evolution. This procedure is followed by sequencing, with the goal of understanding the nature of the mutations which are often found in the binding domains of the phage to its receptors on the target bacteria. Each “trained” mutant is stored separately as a new phage; and the mutant’s ancestor, the “training” method, and the mutations (if detected) are recorded.

## 3. Discussion

Phages are emerging as promising antibacterial agents, mainly in cases of antibiotic-resistant bacteria, antibiotic tolerance, and biofilms [3,15]. Accordingly, in recent years, a steady increase in experimental and clinical phage-related studies has been observed [15]. The high specificity of phages dictates personal-approach treatments in which phages are matched precisely to the target bacteria before use. In our opinion, phages should not be given empirically to the patient without adapting them and testing several aspects beforehand, including the effect of interactions with antibiotics on their efficacy. To this end, it is imperative to have large banks with a variety of phages on the one hand, and rapid and efficient methods for phage isolation on the other. Such a bank should constantly expand by the addition of new bacterial isolates, developing phages against them and storing them ready to be used [6]. Moreover, a worldwide network of such banks will dramatically decrease the chances of the appearance of a threatening difficult-to-treat bacterial epidemic as we see now with viruses. Nevertheless, the situation today indicates quite a difficult and cumbersome process to allocate suitable phages for particular needs. 

The IPB was set up as part of a future phage therapy center (Figure 1) that will have arms of research treatment, phage production under Good Manufacturing Practice (GMP) conditions, and the bank. The bank aims to develop new isolation and storage methods and to have a variety of phages that will serve in research and in response to any bacterial challenge. So far, we have focused on collecting phages mainly against pathogens including the ESKAPE [16]. In parallel, we have also searched for phages against less pathogenic bacteria such as *S. mutans* and other oral and environmental bacteria. These searches are carried out by the lab students and by gifted high school students from the Alpha program of the Hebrew University (https://www.madaney.net/site/programs/alpha/).

To date, the bank contains 289 phages against 21 bacteria species. Their characterization, including full genome sequencing, is a time-consuming and expensive process that is prioritized upon necessity, for instance, if the phages are required for a potential patient. 

In order for this data to be available to the community, we established an initial database within our lab web site (https://ronenhazanlab.wixsite.com/hazanlab/the-404-israeli-phage-bank), which we constantly update, adding the knowledge we accumulate on each phage. These data should also aid in advancing machine learning approaches to utilize phage-bacteria matching bioinformatically.

In summary, our goal is that the IBP will integrate into a worldwide network of similar institutes that will share knowledge and phages in order to coordinate acts against the emergence of multidrug resistant bacteria. 

## Figures and Tables

**Figure 1 antibiotics-09-00269-f001:**
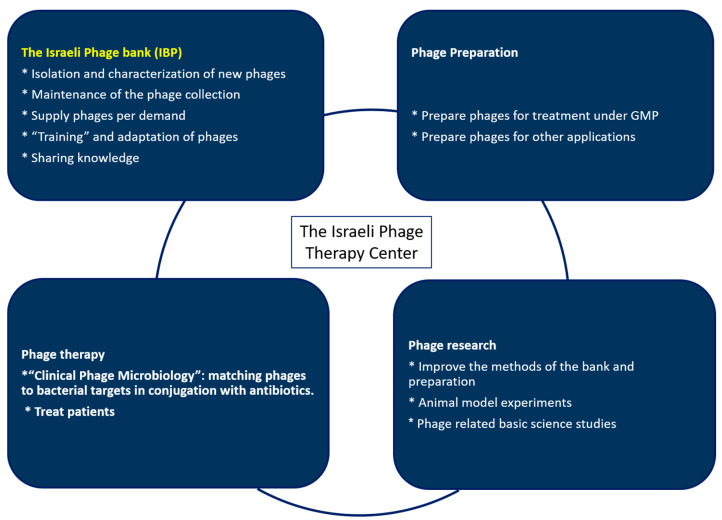
The future phage therapy center’s mode of operation.

**Table 1 antibiotics-09-00269-t001:** Phage collections worldwide.

Phage Collection Name	Number of Phages	Hosts	Link
Adaptive phage therapy (APT) phage bank	~1000	Mainly the ESKAPE pathogens	http://www.aphage.com/the-science/
The Felix d’Hérelle Reference Center for Bacterial Viruses	>400	A few dozen hosts	https://www.phage.ulaval.ca/en/phages-catalog/
The Bacteriophage Bank of Korea	~1000	A few dozen hosts	http://www.phagebank.or.kr/intro/eng_intro.jsp
Leibniz Institute—DSMZ (German Collection of Microorganisms and Cell Cultures)	~300	Unknown	https://www.dsmz.de/collection/collection-experts
ATCC Bacteriophage Collection	~400	A few dozen hosts	https://www.atcc.org/search#q=phage&sort=relevancy&f:productcategoryFacet=[Bacteria%20%26%20Phages]&f:listofapplicationsFacet=[Bacteriophage]
NCTC Bacteriophage Collection	>100	*Streptococcus* ssp *Staphylococcus* ssp *Campylobacter*	https://www.phe-culturecollections.org.uk/products/bacteria/bacteriophages.aspx
Hatfull Lab Phage Collection	>15,000	Species from Actinobacteria phylum	https://phagesdb.org/ http://www.hatfull.org/sea-phages
TUDelft	Unknown	Unknown	https://www.tudelft.nl/en/delft-university-fund/
P.H.A.G.E	Unknown	Unknown	http://www.p-h-a-g-e.org/
Israeli Phage Bank (IPB)	>300	16 different species	https://ronenhazanlab.wixsite.com/hazanlab/the-404-israeli-phage-bank

**Table 2 antibiotics-09-00269-t002:** Target bacteria collection.

Bacteria	Number of Srains
*Acinetobacter baumanii*	12
*Bacillus anthracis*	1
*Burkholderia cepacia*	2
*Burkholderia contaminans*	2
*Burkholderia lata*	1
Others *Burkholderia ssp*	20
*Enterobacter cloacae*	8
*Enterococcus faecalis*	15
*Enterococcus faecium*	10
*Escherichia coli*	8
*Klebsiella pneumoniae*	10
*Mycobacterium abscessus*	6
*Propionibacterium acnes*	36
*Providencia rettgeri*	21
*Providencia stuartii*	20
*Pseudomonas aeruginosa*	120
*Pseudomonas stutzeri*	2
*Pseudomonas syringae*	1
*Salmonella enterica*	2
*Shigella ssp*	30
*Staphylococcus aureus*	40
*Streptococcus mutans*	15
**Total**	**382**

**Table 3 antibiotics-09-00269-t003:** Phage collection.

Bacteria Strain	Clinical Strains	Phages	Percentage of Coverage	Minimum Phages Require for Coverage
*Staphylococcus aureus*	20	25	100%	1
*Klebsiella pneumoniae*	8	30	100%	2
*Pseudomonas aeruginosa*	100	54	100%	9
*Pseudomonas stutzeri*	2	6	100%	1
*Enterobacter spp*	7	20	100%	3
*Burkholderia spp*	23	15	90%	5
*Burkholderia lata*	1	2	100%	1
*Enterococcus faecalis +* *Enterococcus faecium*	8	38	100%	1
*Streptococcus mutans*	15	1	100%	1
*Acinetobacter baumannii*	8	19	100%	2
*Propionibacterium acnes*	36	22	86%	1
*Providencia spp*	35	10	100%	3
*Escherichia coli*	5	25	100%	1
*Bacillus anthracis*	1	6	100%	1
*Salmonella*	30	1	100%	1
*Shigella*	2	12	100%	1
*Mycobacterium abscessus*	6	3	100%	1
**Total**	**307**	**289**	**97%**	**35**

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
