# Peer review of "The Israeli Phage Bank (IPB)"

_antibiotics, 2020, doi:10.3390/antibiotics9050269_

Round 1

Reviewer 1 Report

IPB has been well introduced with overall plan. However, more specific discussion should be needed for the phages as alternatives over antibiotics. This might be asked as “How does IPB deal with the antibiotic resistance in bacteria?".

  1. For the therapeutic use of phages and clinical trials, the purification is one of the important steps. So, the purification should be stated in the Method section.
  2. Phage-binding receptors are another important factor for the phage-based control and detection as well. So, the alteration in the phage-binding receptors on the surface of antibiotic-resistant pathogens should be addressed here.
  3. Phage mutants should be mentioned and specifically state how the phage mutant is considered in IPB.

Author Response

We thank the reviewer for the constructive comments. Please see our answers below in bold:

IPB has been well introduced with overall plan. However, more specific discussion should be needed for the phages as alternatives over antibiotics. This might be asked as “How does IPB deal with the antibiotic resistance in bacteria?".

> We thank the reviewer for drawing our attention to the fact that in this manuscript, we did not emphasize that in our view, phages are not alternatives to antibiotics but an additional weapon in the arsenal against bacteria. Our pipeline, which is submitted elsewhere (Gelman et.al, final steps of preparations), is always to test antibiotics, phages and combinations of them both. This point was clarified in lines 17, 33-34 and 152-154 of the revised manuscript.

For the therapeutic use of phages and clinical trials, the purification is one of the important steps. So, the purification should be stated in the Method section.

> So far we do not have a GMP purification ability suitable for human treatment. Therefore, although the phages which we used for treatment were isolated and characterized by us, the purification was outsourced, mainly to APT.

Phage-binding receptors are another important factor for the phage-based control and detection as well. So, the alteration in the phage-binding receptors on the surface of antibiotic-resistant pathogens should be addressed here.

> When we “train” our phages, we do it using the classical Darwinian’s mutation-selection methods that served humanity all along history to domesticate animals, plants and microorganisms. After we get an improved phage on the target bacteria, we sequence and test which mutation caused the improvement. In many cases the mutations are in unknown proteins and we do not know if these are recognition factors. Furthermore, not always are the phage-binding receptors known. We referred to this point in line 146-149 of the revised manuscript.

Phage mutants should be mentioned and specifically state how the phage mutant is considered in IPB.

> Phage mutants are considered as “new phages” in the sense that they are stored and recorded as independent phages, albeit with a comment about their ancestor strain, method of creation and their mutations, when detected. We added that to lines 149-150 in the revised manuscript.

Reviewer 2 Report

The development of phage banks from around the world are much needed. Although this collection at the Israeli Phage bank is still in its infancy, its continuation will be of significant value. The design of the facility seems to have been well thought out and particular care has been taken to preserve phage lysates as well as phage DNA inside the host.

One thing I would like to address is whether the facility is also archiving concentrated viral preps (prior to amplification with clinical strains). In the event that a clinical sample is negative for phage infection, these preps can then be used to hunt for additional phages without needing to source additional environmental samples. Additionally, such samples could also potentially be used as a sources for contact tracing of emerging pathogens. 

In terms of layout, I find it difficult to read tables which are split over 2 pages (suggest landscape view for Table 1?). The white text of figure 1 needs to be larger. 

Table 3 is a bit misleading in terms of % coverage. For example in having only 8 clinical strains of K. pneumoniae, you should reach 100% coverage with a total of 8 phages. As presented the data implies that you need all 30 phages to achieve 100% inclusivity. I would suggest the inclusion of an additional column which communicates the minimum number of phages required to hit 100% coverage. 

Line 120 reads Adoptive not Adaptive

Author Response

We thank the reviewer for the constructive comments. Please see our answers below in bold:

The development of phage banks from around the world are much needed. Although this collection at the Israeli Phage bank is still in its infancy, its continuation will be of significant value. The design of the facility seems to have been well thought out and particular care has been taken to preserve phage lysates as well as phage DNA inside the host.

> We thank the reviewer for recognizing the importance of phage banks.

One thing I would like to address is whether the facility is also archiving concentrated viral preps (prior to amplification with clinical strains). In the event that a clinical sample is negative for phage infection, these preps can then be used to hunt for additional phages without needing to source additional environmental samples. Additionally, such samples could also potentially be used as a sources for contact tracing of emerging pathogens.

> We do store the samples at 40 C after the initial use. Fecal samples are stored for 2 months and environmental samples (soil and water) are stored longer. We re-use them routinely in screens against target bacteria. We added that to lines 89-90 in the revised manuscript.

In terms of layout, I find it difficult to read tables which are split over 2 pages (suggest landscape view for Table 1?). The white text of figure 1 needs to be larger.

> Corrected

Table 3 is a bit misleading in terms of % coverage. For example, in having only 8 clinical strains of K. pneumoniae, you should reach 100% coverage with a total of 8 phages. As presented the data implies that you need all 30 phages to achieve 100% inclusivity. I would suggest the inclusion of an additional column which communicates the minimum number of phages required to hit 100% coverage.

>

Line 120 reads Adoptive not Adaptive

> Corrected

Round 2

Reviewer 1 Report

The manuscript has been well revised according to the comments.